# A New Approach for Improving the Antibacterial and Tumor Cytotoxic Activities of Pipemidic Acid by Including It in Trimethyl-β-cyclodextrin

**DOI:** 10.3390/ijms20020416

**Published:** 2019-01-18

**Authors:** Margherita Lavorgna, Rosa Iacovino, Chiara Russo, Cristina Di Donato, Concetta Piscitelli, Marina Isidori

**Affiliations:** Dipartimento di Scienze e Tecnologie Ambientali, Biologiche e Farmaceutiche, Università della Campania “L. Vanvitelli”, Via Vivaldi 43, I-81100 Caserta, Italy; margherita.lavorgna@unicampania.it (M.L.); rosa.iacovino@unicampania.it (R.I.); chiara.russo@unicampania.it (C.R.); cristina.didonato@unicampania.it (C.D.D.); concetta.piscitelli@unicampania.it (C.P.)

**Keywords:** trimethyl-β-cyclodextrin (TRIMEB), pipemidic acid, inclusion complex, microbial activity, antitumoral activity

## Abstract

Pipemidic acid (HPPA) is a quinolone antibacterial agent used mostly to treat gram-negative infections of the urinary tract, but its therapeutic use is limited because of its low solubility. Thus, to improve drug solubility, natural cyclodextrins (CDs) are used for their ability of including guest molecules within their cavities. The aim of this work was to evaluate the antibacterial activity and the preliminary anticancer activity of HPPA included into Heptakis (2,3,6-tri-*O-*methyl)-β-cyclodextrin (TRIMEB) as a possible approach for a new innovative formulation. The inclusion complex of HPPA with TRIMEB was prepared in solid state by the kneading method and confirmed by FT-IR and powered X-ray diffraction. The association in aqueous solutions of pipemidic acid with TRIMEB was investigated by UV-Vis spectroscopy. Job’s plots have been drawn by UV-visible spectroscopy to confirm the 1:1 stoichiometry of the host–guest assembly. The antibacterial activity of HPPA, TRIMEB and of their complex was tested on *Escherichia coli*, *Pseudomonas aeruginosa*, and *Staphilococcus aureus*. The complex was able to increase 47.36% of the median antibacterial activity of the free HPPA against *E. coli* (IC_50_ = 249 µM vs. 473 µM). Furthermore, these samples were tested on HepG-2 and MCF-7. After 72 h, the median tumoral cytotoxicity exerted by the complex was increased by 78.08% and 94.27% for HepG-2 and MCF-7 respectively, showing a stronger bioactivity of the complex than the single HAPPA.

## 1. Introduction

The discoveries about the quinolone class of antimicrobial agents evolved along the years. From the accidental discovery of nalidixic acid to the last generation of quinolones, over 50 years have passed and their use has still a clinical effectiveness.

Many infections are treated with quinolones exploiting their ability to inhibit the bacterial DNA topoisomerase II and IV, and the knowledge of their molecular mechanism of action has led to the development of new quinolones with improved effects against bacterial topoisomerases [1,2,3]. Quinolones are molecules produced by synthesis showing, in their evolution, differences in activity because of various substituents added to the quinolone nucleus. In fact, these antimicrobial agents have been classified from the first-generation to the fourth-generation quinolones going from molecules effective against aerobic gram-negative bacteria to molecules with an improved activity against gram-positive bacteria and anaerobes. Despite their unfavorable physico-chemical profiles, quinolones might have good opportunities to offer new compounds, especially selecting molecules with greater activity against infections difficult to treat and with lesser patient adverse reactions; although, at the moment, the main concern for the clinical use of quinolones is the bacterial resistance to these agents.

In the last years, quinolones have been investigated for their application as anticancer drugs since their eukaryotic cytotoxicity was demonstrated, due to their affinity for the DNA of eukaryotic topoisomerases [1,4]. Quinolones could have some advantages in the clinic over the conventionally employed topo II drugs, such as etoposide and doxorubicin. They are quite safe, without any significant cardiotoxicity, and easily cross eukaryotic and prokaryotic cell membranes, and being synthesized, new analogs might be easily produced [5].

Although the investigation might continue in the development of innovative molecules, it is interesting to study already known quinolones utilizing systems able to improve their delivery as well as their bioavailability with effects in reducing effective drug doses and positive results in decreasing the few adverse events in treated patients. In this regard, cyclodextrins have been extensively investigated to modify the properties of drugs [6,7]. Cyclodextrins (CDs) are cyclic oligosaccharides consisting of six (α-CD), seven (β-CD), or eight (γ-CD) units of D-glucopyranose in chair conformations, produced, as well-known, from enzymatic degradation of the linear amylase component of starch. The structure truncated cone of these natural CDs is relatively rigid in terms of size and shape, the molecular arrangement is a result of the free rotation around the glycosidic bonds [8]. They are moderately water soluble due to the hydroxyl groups in the cavity rims [9]. It is generally accepted that CDs can form an inclusion complex in aqueous solutions, and a lipophilic guest molecule or moiety may relocate to the inner cavity [10]. Usually, the formation of inclusion complex involves noncovalent host–guest interactions such as electrostatic, van der Waals, hydrophobic and/or hydrogen bonding [9,11,12]. The lipophilic cavity of CDs provides a microenvironment into which appropriate size nonpolar molecules or molecular fragments can enter to form inclusion complexes. Due to this special property, CDs are extensively used in clinic to improve the solubility of drugs, reducing bitterness, enhancing stability, and decreasing tissue irritation [13,14].

Methylation of the primary and secondary hydroxyl groups increases the hydrophobic cyclodextrinic cavity and alters its solubility in water and organic solvents [9]. So methylated CDs have already been used in several applications, for example, in the drug delivery to control the drug biodistribution, reducing side effects, and to improve bioavailability [11]. In particular, modified cyclodextrins like Heptakis (2,3,6-tri-*O*-methyl)-β-cyclodextrin (TRIMEB) are best known to be more soluble with respect to natural CDs [15,16] and for this reason, their use in pharmaceutical formulation is desirable [17].

In this context, similar to a previous study of Iacovino et al., 2013 [18] who evaluated the antibacterial activity and the tumoral cytotoxicity of the pipemidic acid (HPPA) complexed with the natural β-CD, we studied the antimicrobial and the antitumoral activities of HPPA complexed with TRIMEB. Thus, this study was aimed to recycle this “old” drug, proposing a new possible approach for an innovative therapeutic formulation. Pipemidic acid (Figure 1), 8-ethyl-5,8-dihydro-5-oxo-2-(1-piperazinyl)-pyrido(2,3-*d*)pyrimidine-6-carboxylic acid [19] is a first-generation quinolone used as a therapeutic agent for urinary tract infection. It is especially active against aerobic, gram-negative bacteria but not very active against aerobic, gram-positive bacteria or anaerobic bacteria.

The aim of this present study was to evaluate the biological activities of HPPA:TRIMEB. This inclusion complex was prepared in the solid state, and its formation was confirmed by FT-IR Spectroscopy and X-ray Powder Diffractometry, while the association in aqueous solutions of pipemidic acid with TRIMEB was investigated by UV-Vis spectroscopy.

Furthermore, in order to evaluate the possible differences between HPPA and HPPA:TRIMEB biological activities, different assays were performed. The antibacterial activity of both HPPA and the complex HPPA:TRIMEB was evaluated by the Microbial Susceptibility Assay on three bacterial species implicated in human infections—two gram-negative bacteria, *Escherichia coli* ATCC 13762 and *Pseudomonas aeruginosa* ATCC 9027, and one gram-positive bacterium, *Staphylococcus aureus* ATCC 6538—determining the effective median inhibitory bacterial growth concentration (IC_50_). Furthermore, to evaluate the different antitumor potency between the HPPA and the inclusion complex, both were tested on human hepatoblastoma HepG-2 and on MCF-7 cell lines of breast adenocarcinoma by the MTT assay, evaluating the median inhibition concentration (IC_50_).

## 2. Materials and Methods

### 2.1. Reagents

TRIMEB (CAS 55216-11-0) and HPPA (CAS 51940-44-4) were purchased from Sigma-Aldrich (St. Louis, MO, USA) (Figure 1).

### 2.2. Preparation of the Solid Binary System

The physical mixture (PM) between the HPPA and TRIMEB was prepared by adding 0.047 g of TRIMEB with 0.010 g of HPPA and mixing them in a mortar.

The HPPA:TRIMEB solid binary system was prepared in 1:1 molar ratio by the kneading method. The kneading product (KND) was obtained by adding a small volume of a water–methanol (50/50, *v*/*v*) solution to the HPPA (0.020 g) and TRIMEB (0.094 g), and the resulted mixture was homogeneously pasted until the solvent was completely removed. The sample was dried at 40 °C in the oven for 24 h to remove traces of the solvent. The dried mass was pulverized.

### 2.3. Fourier Transform Infrared (FT-IR) Spectroscopy

The FT-IR spectra, for the samples of the HPPA, TRIMEB, physical mixture and for the kneading product were recorded using a FT-IR Perkin Elmer Spectrum GX spectrometer (Waltham, MA, USA). The analysis was carried out using the KBr pellet technique, and the tablets were prepared by compressing the powder. The spectral range was kept from 4000 to 400 cm^−1^, with a resolution of 1 cm^−1^.

### 2.4. X-ray Powder Diffraction (XRD)

XRD diffraction tests were conducted using a Bruker AXS D8 Advance diffractometer (Karlsruhe, Germany) with a tube anode Cu and a graphite monochromator. The analysis was performed at room temperature and at 40 kV and 30 mA. The diffractograms were recorded in the 2θ angle range between 3° and 35° and process parameters with scanning speed 0.01 θ/s

### 2.5. The Job Plot Method for the Determination of Stoichiometry

The continuous variation method or Job method [20,21] was utilized for the determination of the stoichiometry of the complex. In agreement with this method, 0.05 mM unbuffered solutions of HPPA and a solution of TRIMEB at the same concentration were mixed at different molar ratios R = (HPPA)/((HPPA)(TRIMEB)) maintaining the volume constant. For each complex, the stoichiometric ratio was obtained by reporting (ΔA × R) against R (where ΔA is the difference in absorbance of the drug in the absence and in the presence of the CD); the R value corresponds to the maximum of the curve obtained. All measurements were recorded in the wavelength range 200–400 nm at room temperature. For all UV-Vis spectroscopy studies, a UV-1700 Spectrometer (Shimadzu, Tokyo, Japan) was used with a 1 cm quartz cuvette.

### 2.6. Ultraviolet-Visible (UV-Vis) Spectroscopy

The molar ratio titration method was used to estimate the binding constants for all the complexes under investigation [22,23]. TRIMEB solutions at varied concentration (from 0 to 1.85 mM) were added to buffered and unbuffered solutions of HPPA at constant concentration (0.05 mM). The absorbances of each obtained solution was measured at different pHs. As the evaluation of Kb by direct spectroscopic methods relies on analytical differences between the free and the complexed drug [24], changes in the absorption intensity of HPPA was monitored as a function of the cyclodextrin concentrations. The maximum absorption wavelength of HPPA was found at 323.5nm for pH = 4.3, at 326.0 nm for pH = 5.3, and at 332.0 nm for pH = 8.3.

All absorption measurements were made against a blank solution treated in the same way. To conveniently calculate the Kb, we rearranged the Benesi–Hildebrand equation [22] into the straight line form [25] shown in Equation (1).
(1)$$A=-\frac{1}{Kb}\frac{A-A_{0}}{\left[H\right]}+A_{0}+∆\varepsilon \left[G\right]$$
where *A* and *A*_0_ are the absorbance of HPPA in the presence and absence of cyclodextrin, respectively, *Kb* is the stability constant, [*H*] and [*G*] are the concentrations of TRIMEB and HPPA, respectively, and Δ*ε* is the difference in the molar absorptivities between the free and complexed guests.

### 2.7. Bioactivity Evaluation

#### 2.7.1. Microbial Susceptibility Test

Bacteria were stored at −80 °C in 90% (*v*/*v*) glycerol in Tryptic Soy Broth, Soybean-Casein Digest Medium (TSB, Oxoid, Milano, Italy) until the overnight culture in TSB at 37 °C. This broth-based turbidometric assay was performed under sterile conditions, using 96-well plates (Sarstedt, Italy). First, 100 μL of TSB broth medium was added to the wells. Then, 100 μL of HPPA, TRIMEB, or the HPPA:TRIMEB complex was added in sextuplicate in the sample wells at the desired concentrations (chosen after range finding tests), while the physiologic solution (0.9% NaCl) was added into the negative control wells. HPPA was tested from 0.04 to 16.48 mM, while the complex was assessed from 0.001 up to 2.88 mM. Furthermore, in order to exclude the potential inhibition effect of the single TRIMEB, it was tested up to the highest concentration of 3.49 mM. In each well, except for the blank control (with only TSB medium), 100 μL of the bacterial culture corresponding to 1 × 10^4^ CFU [26] was inoculated. The plates were covered by a sterile film to avoid evaporation and were incubated at 37 °C. After 24 h, the optical density of each well was recorded at 620 nm, using a microplate reader (Spectra Fluor, Tecan, Mannedorf, Switzeland). Three independent experiments were performed. A positive control was carried out using streptomycin at the starting concentration of 10 μg/mL. The bacterial inhibition percentage (IC%) was determined as follows:
IC% = 1 − [(OD_620_ of the test sample − OD_620_ of the blank)/(OD_620_ of the negative control − OD_620_ of the blank)] × 100(2)

Moreover, the effective percentages obtained from the three independent experiments were pooled using Prism5 (Graphpad Inc., San Diego, CA, USA) to estimate the concentrations giving 50% microbial growth inhibition (IC_50_) by non-linear regression (log agonist vs. normalized response-variable slope). ANOVA and Dunnett’s multiple comparison test estimated the significant differences among the single HPPA, the single TRIMEB, and the complex.

#### 2.7.2. MTT-Assay

The preliminary anticancer activity of HPPA, TRIMEB, and HPPA:TRIMEB was assessed on human hepatoblastoma HepG-2 and on breast adenocarcinoma MCF-7 cell lines by the colorimetric MTT (3-(4, 5-dimethylthiazol-2-yl)-2, 5 diphenyl tetrazolium bromide) assay according to Baharum et al. [27] with slight modifications. Cells were maintained in Roswell Park Memorial Institute (RPMI) 1640 (supplemented by 10% Fetal Bovine Serum (FBS), 2% of HEPES and l-glutamine, and 1% of penicillin/streptomycin) and cultured in tissue culture flasks at 37 °C in a 95% air/5% CO_2_ incubator under saturating humidity. After reaching the 80%–90% confluence, cells were collected and counted with the vital dye, trypan blue, using an optical microscope. A total of 10^4^ cells/well were seeded in 100 µL/well of Dulbecco’s Modified Eagle’s Medium (DMEM) supplemented by FBS and antibiotics in 96-well microplates in quadruplicate and were incubated at 37 °C for 24 h. Then, the DMEM was removed and 200 µL of samples diluted in fresh medium was added to each well and incubated for 24, 48, and 72 h at 37 °C. Sample concentrations were chosen after some range finding tests, so that HPPA was tested from 0.026 to 16.48 mM, while HPPA:TRIMEB was tested from 0.009 to 1.44 mM. In addition, to exclude the potential cytotoxic effect of the single TRIMEB, it was tested at the highest concentration of 1.75 mM. Negative control wells contained only DMEM.

After the incubation time, 20 µL of yellow MTT solution (5 mg/mL) was added to each well and the plates were incubated for a further 4 h at 37 °C in order to have in mitochondria the reduction of MTT salts in formazan crystals which were dissolved with 100 µL of 2-propanol. The absorbance was recorded at 590 nm using an Ultra Multifunctional Microplate Reader (TECAN). The cell inhibition percentage was evaluated according to the following equation:
IC% = 1 − [OD_590_ of the test sample/OD_590_ of the negative control] × 100(3)

The three independent experiments were pooled and statistically analyzed by Prism 5 (GraphPad Inc., San Diego, CA, USA). IC50 values (concentrations causing the 50% inhibition of cell growth) were calculated by nonlinear concentration/response regression model. ANOVA and Dunnett’s multiple comparisons test were used to estimate the significant differences among HPPA, TRIMEB, and their complex.

## 3. Results and Discussion

### 3.1. FT-IR Spectroscopy

The initial assessment of the inclusion of HPPA into TRIMEB is based on the FT-IR spectroscopy studies. The FT-IR spectrum gives detailed information about the functional groups involved in the interaction when the complex is formed. The spectra of TRIMEB, HPPA:TRIMEB, products obtained by the kneading techniques (KND), physical mixing product (PM), and HPPA are shown in Figure 2.

The FT-IR spectrum of TRIMEB showed prominent absorption bands at 2933 cm^−1^ (symmetric and asymmetric C–H stretching), 1158 cm^−1^ (for C–O stretching vibrations), and 1016 cm^−1^ near the region 1076–1022 cm^−1^ (C–O–C stretching vibrations), that are in good agreement with the values reported by Rizzi et al. [28]. The band at 3300 cm^−1^ (–OH stretching vibrations) characteristic of β-CD [29] was not observed due to the methylation of the –OH moieties. In the FT-IR spectrum of HPPA, characteristic bands are shown to be correlated to the stretching vibration of the –OH group, the intramolecular hydrogen bonding at 3046 cm^−1^, the –NH stretching vibration of the imino-moiety of piperazinyl groups at 3366 cm^−1^, the carbonyl C=O stretching at 1721 cm^−1^, and the bending vibration of O–H group at 1261 cm^−1^. The peak at 1627 cm^−1^ was assigned to the N–H bending vibration of the quinolones [30]. The spectrum of the PM product shows approximate superimposition of the individual patterns of both TRIMEB and HPPA. In the spectrum of the KND, the –NH stretching region of HPPA (3366 cm^−1^) is shifted at 3469 cm^−1^. The strong carboxyl carbonyl stretching vibration peak at 1721 cm^−1^ of HPPA disappeared, and the bending vibration of O–H group of carboxylic acid at 1261 cm^−1^ of HPPA is shifted to 1252 cm^−1^ in the complex, indicating the dissociation of the intermolecular hydrogen bonding and interaction through hydrogen bonding with the cyclodextrin.

### 3.2. Powder X-ray Diffraction

XRD diffraction studies are useful to allow the identification of the inclusion complex based on the fact that the crystallinity of the compounds changes upon host–guest interaction. XRD patterns of pure HPPA, TRIMEB, their physical mixture (PM), and their kneading product (KND) are shown in Figure 3. The powder diffractogram for the HPPA presents diffraction angles of 2θ at 11.32°, 14.68°, 19.07°, 21.03°, 24.83°, and 34.16° according to literature data [31,32]. The XRD patterns of TRIMEB revealed several diffraction peaks indicative of its crystalline character [33]. For the kneading products (HPPA:TRIMEB), the observed diffraction pattern reveals their amorphous character [34] while the characteristic peaks of pure HPPA and TRIMEB are still present in the diffractogram of the physical mixture, although with a reduced intensity.

### 3.3. UV-Vis Spectroscopy

The inclusion of the guest molecule inside the cyclodextrinic cavity can modify its microenvironment; as a consequence, the UV-Vis spectrum of the complexed molecule shows changes of the absorbance [35,36]. The 1:1 stoichiometry of the complex was determined using the Job method [20], considering that the maximum of the curve is at *R* = 0.5 (Figure 4). In an aqueous solution, the HPPA exists in different forms depending on the pH value and each of them may form complexes with cyclodextrins [18,37].

Thus, the inclusion of HPPA with TRIMEB was studied in unbuffered (pH = 5.3), sodium acetate buffered (pH = 4.3), and Tris HCl buffered (pH = 8.3) solutions. The results of the dependence of the HPPA absorbance on the TRIMEB concentration are shown in Figure 5. The maximum absorption wavelength of HPPA was pH dependent, being 323.5 nm at pH 4.3, 326.0 nm at pH 5.3, and 332.0 nm at pH 8.3. These results suggest that the inclusion complex was formed between TRIMEB and HPPA. The *K*b can be obtained from the absorbance data using the modified Benesi–Hildebrand Equation (1) reported in the Materials and methods section [22,25].

Therefore, a plot of A versus (*A* − *A*_0_)/(*TRIMEB*), should give a straight line with slope −1/*K*b. The calculated stability constants at different pH were listed in Table 1, from which the formation constant values were very sensitive to pH: *K*b8.3 > *K*b5.3 > *K*b4.3. The obtained data suggested that the interaction is favored in basic media.

Several factors contribute to the stability of the host–guest complex such as the intramolecular hydrogen bonds between the –OH groups of the glucuriranose unit, the Coulomb electrostatic forces, and the van der Waals interactions. The TRIMEB, as a consequence of the substitution of its hydroxyl groups, cannot form hydrogen bonds; the stability of the complex reveals the importance of the hydrophobic character of macrocyclics.

Comparing our data with those already reported in Reference [18], TRIMEB showed a selective binding for the negative charged HPPA with a more hydrophilic character compared to the same drug when complexed with β-CD in the same conditions. For the cation form of the HPPA, the binding constant is lower for the complex with TRIMEB compared to that with β-CD because no intramolecular hydrogen bonds are formed. 

### 3.4. Microbial Susceptibility Test

The potential antimicrobial activities exerted by HPPA, TRIMEB, and HPPA:TRIMEB against two gram-negatives, *E. coli* and *P. aeruginosa*, and one gram-positive, *S. aureus*, after 24 h of incubation at 37 °C were expressed as the effective concentrations inhibiting 50% of the bacterial growth (IC50s), with 95% confidence limits. The antimicrobial activity results are reported in Table 2.

No effect was observed for the single TRIMEB against all bacteria strains up to the highest concentration tested equal to 3497 µM. This not toxic bacterial activity of the modified-CD TRIMEB was observed also by Bar and Ulitzur [38]. Before starting HPPA:TRIMEB experiments, we tested the single antibiotic agent, obtaining results in the same confidence interval ranges of our previous outcomes [18]. In the present study, among the strains tested, both HPPA and HPPA:TRIMEB showed their highest activity on *P. aeruginosa*, with IC_50_ values equal to 98 and 122 µM respectively, but no significant difference was found between them, as reported also by Iacovino and coauthors [18], where the pipemidic acid was tested in complex with the natural CD. To the best of our knowledge, no studies were performed testing single TRIMEB or complexed with HPPA on *P. aeruginosa* and not many recent studies have been conducted exposing *P. aeruginosa* to pipemidic acid. Some old studies were performed in 1975 by Shimizu et al. [39], which found the Minimal Inhibitory Concentration (MIC) of this quinolone at 25 µg/mL (82 µM) on *P. aeruginosa* no.12, while in 1987, the pipemidic acid MIC value was found equal to 12.5 µg/mL (41 µM) on the strain *P. aeruginosa* PA04009. 

Regarding *E. coli*, HPPA was able to inhibit 50% of the bacterial growth at the concentration of 473 µM, while its complex was able to cause the same median growth-inhibitory effect at the concentration equal to 249 µM (Table 2), with statistical significant difference from the single HPPA (*p* < 0.01, Dunnett’s test) and a reduction in the percentage of the single HPPA in the complex equal to 47.36% as depicted in Figure 6. 

Certainly, these results could be of great interest in human therapy because HPPA:TRIMEB could be used to reduce the amount of drugs needed to inhibit the growth of *E. coli* implicated in urinary tract infections for which HPPA generally is used [40]. Furthermore, comparing the antimicrobial activities of both the HPPA:TRIMEB (here studied) and the HPPA:β-CD (evaluated in our previous study, Iacovino et al. [18]), we can conclude that the complex with the TRIMEB is the most active on *E. coli.* In fact, when HPPA was hosted by β-CD, the median antimicrobial effect was reached with a HPPA concentration reduction of 25.93% compared to the single HPPA concentration; when HPPA is complexed by TRIMEB, the median growth inhibition on *E. coli* is reached with a further concentration reduction of 21.43% and a total reduction equal to 47.36% compared to the use of the single HPPA (Figure 6 and Table 2).

In order to have a wider overview of the growth inhibition percentage trend of both HPPA and HPPA:TRIMEB in all bacteria strains, the effect–concentration dependent curves are reported in Figure 7, underlining, especially in *E. coli*, the increase of the growth inhibition from the single acid to the complex, still at the lowest tested concentrations. 

In *E. coli*, the HPPA Lowest Observed Effect Concentration (LOEC), was equal to 82 µM, while for the complex, it was equal to 72 µM. Our results, regarding the effectiveness of the complexation of the pipemidic acid, could be compared to those of Yang et al. [41], who, complexing the acid with some transition metals, observed a notable activity of the complex in *E. coli*, a weak activity against *P. aeruginosa*, and no effect on *S. aureus*. These authors suppose that the entrance of the drug into the bacterial cells improves its action with DNA in vivo. In addition, the capacity of molecular complexes having TRIMEB as a host to improve the antimicrobial properties of the compounds is also confirmed by Marques et al. [42] and Ramos et al. [43]. Probably, the intimate contact between CD complexes and microrganisms enhances the transfer of the compounds to the bacterial cell and, after the bacterial degradation, the release of the compounds due to the CD-saccharides (Trojan-horse mechanism), as suggested by Jaiswal et al. [44].

### 3.5. MTT-Assay

The potential anticancer activity of HPPA, HPPA:TRIMEB, and single TRIMEB was expressed as IC50, the concentration inhibiting 50% of the MCF-7 and HepG-2 viability, after 24, 48, and 72 h of exposure. No cytotoxic effect was observed testing single TRIMEB on both cell lines up to the highest concentration tested (1750 µM), different from the effect shown on murine endothelial cells and on human colorectal cells [45,46]. HPPA and HPPA:TRIMEB cytotoxicity outcomes are reported in Table 3. An evident cytotoxic effect of HPPA:TRIMEB was observed in HepG-2, where a significant (*p* < 0.01) inhibition of the cell growth was observed after 48 and 72 h when compared to the single antibiotic agent (Dunnett’s test). Indeed, after 72 h of exposure, IC50 values were equal to 260 and 57 µM for single HPPA and the complex, respectively, with a reduction of the concentration of HPPA causing the median cell growth inhibition equal to 78.08% when complexed. In Figure 8, the different activities found for the HPPA:TRIMEB complex (here studied) and the HPPA:β-CD complex [18] are depicted, and it is clear that the complex with β-CD, showing a reduction of HPPA percentage equal to 53.33%, is less active compared to the complex with TRIMEB.

On MCF-7, HPPA was less active than the complex, showing IC_50_ values in the order of thousand or several hundreds of µmol/L (Table 3, Figure 9).

On the contrary, when the HPPA:TRIMEB was tested on the breast cancer cells, the inhibition of the cell growth was shown at 1025, 160, and 43 µmol/L after 24, 48, and 72 h of exposure, respectively, with a statistical significant difference from the single acid. After a 72 h exposure, the concentration of HPPA causing 50% of the cell viability inhibition was 94.27% less than the single HPPA (Figure 8). Also on MCF-7, the HPPA:TRIMEB was more active than the complex with β-CD, for which the concentration of HPPA after 72 h exposure was able to reduce the cell growth by 82.95%.

The enhanced efficacy of chemotherapeutic agents in solid tumor cells by methylated cyclodextrins was confirmed by Mohammad and coauthors in 2015 [47], who combined the antineoplastic drug Doxorubicin with the methyl-β-cyclodextrin. These authors observed a potentiated doxorubicin-induced cytotoxicity in both human adenocarcinoma breast cancer cells and murine hepatoma cells due not only to the effect of the methyl-β-cyclodextrin on the intracellular uptake of the single drug but also to the complex able to induce apoptotic death and an involvement of p53 and Fas receptor ligand complex, indicative of interactions between the two molecules. Moreover, the cytotoxic effect of antibiotic agents such as quinolones complexed with cyclodextrins as powerful antitumoral tool was confirmed by Murugan et al. [48] who evaluated the anticancer activity of the host–guest complexation between amodiaquine and native cyclodexstrins.

## 4. Conclusions

The constant development of drug delivery systems based on carriers such as cyclodextrins has been a good strategy to improve the therapy efficacy by decreasing the drug active dose and the subsequent adverse effects. This study showed that the complex HPPA:TRIMEB enables significantly the antibacterial activity of HPPA against *E. coli*. The use of cyclodextrins strongly reduces the amount of the antimicrobial guest to obtain the inhibition of the bacterial growth contributing to the decrease of its toxic potential causing adverse events in treated patients and, in particular, to its relative susceptibility to the development of bacterial resistance.

Furthermore, HPPA complexed with TRIMEB could be a promising antitumoral agent since the results on the two cell lines, herein tested, definitely show that the complex has a higher cytotoxicity than the free HPPA, and it could be considered for future investigations against other human carcinoma cell lines. In conclusion, future potential uses for the old and new quinolones, in addition to their confirmed efficacy as antibacterial agents, is feasible, especially when they are complexed in natural or derivatives cyclodextrins, as shown here with the pipemidic acid, a first generation quinolone.

## Figures and Tables

**Figure 1 ijms-20-00416-f001:**
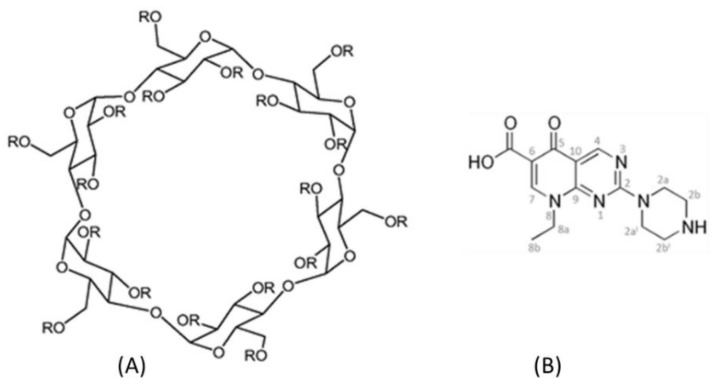
The molecular structure of Heptakis (2,3,6-tri-*O*-methyl)-β-cyclodextrin (TRIMEB), R = Me. (**A**) and pipemidic acid (HPPA) (**B**).

**Figure 2 ijms-20-00416-f002:**
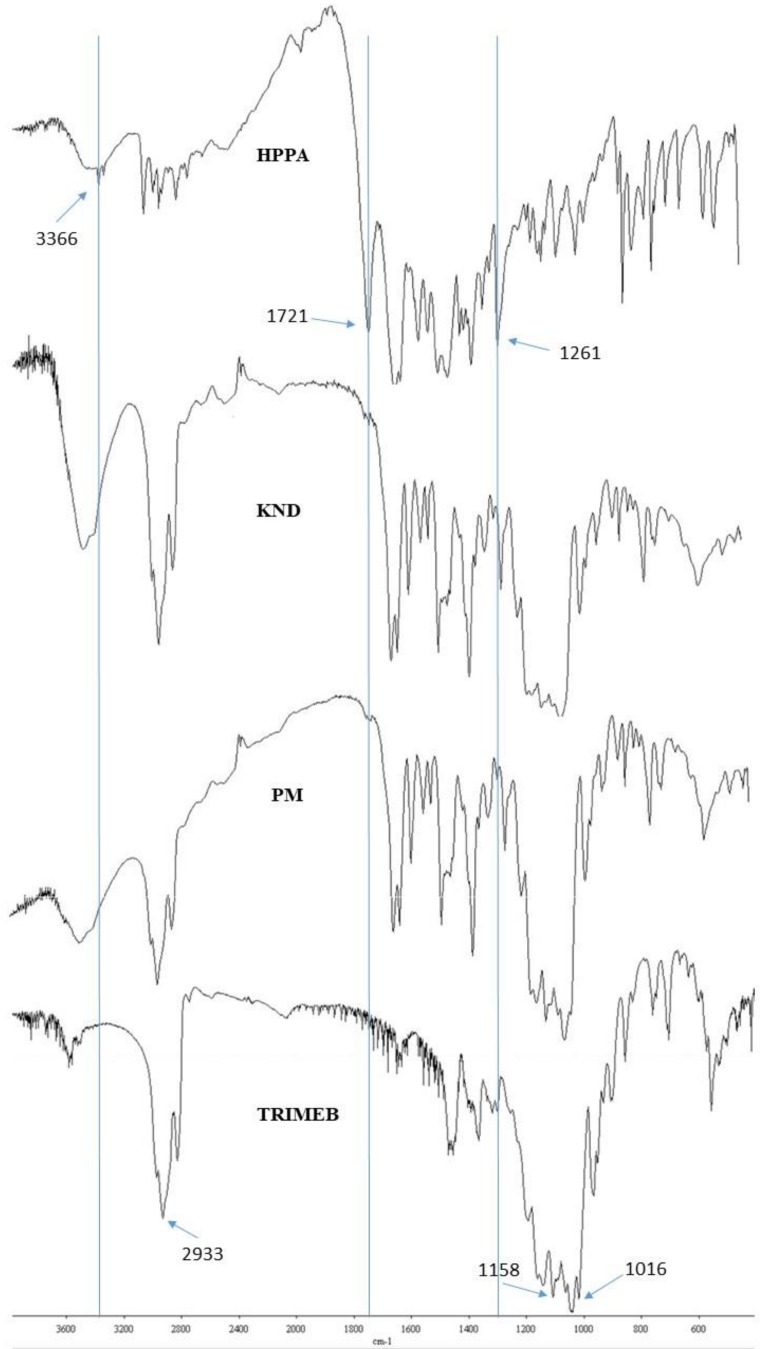
FT-IR spectra of TRIMEB, the kneading product (KND), the physical mixing product (PM), and HPPA.

**Figure 3 ijms-20-00416-f003:**
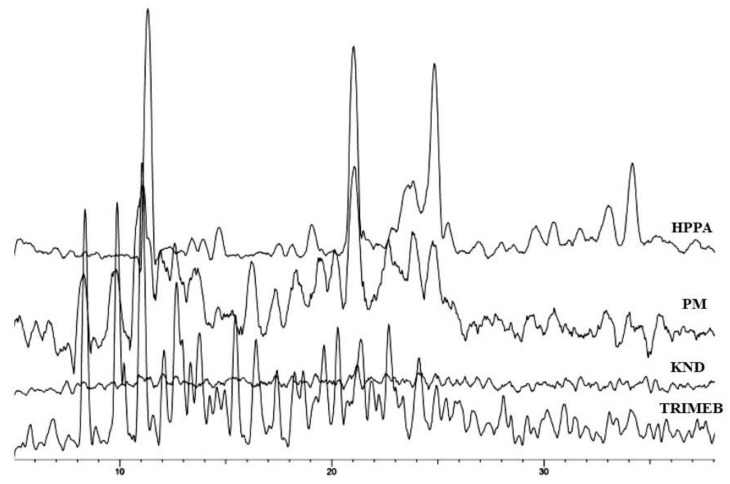
The XRD patterns of HPPA, TRIMEB, KND, and PM.

**Figure 4 ijms-20-00416-f004:**
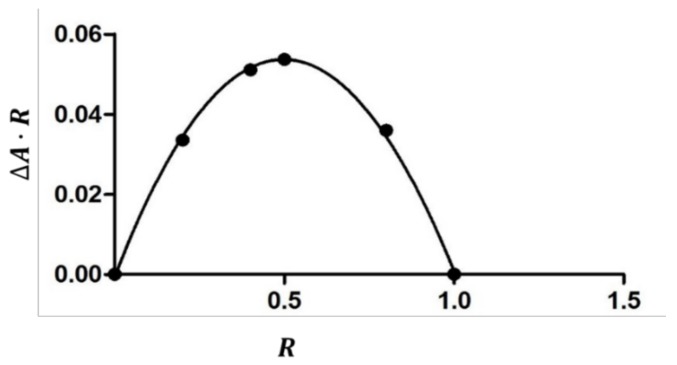
The job plot for the complex HPPA:TRIMEB at λ =324 nm.

**Figure 5 ijms-20-00416-f005:**
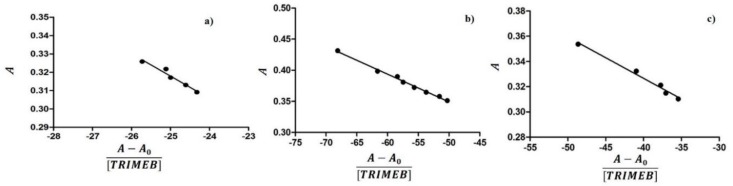
The dependence of the HPPA absorbance from the TRIMEB concentration in aqueous solutions at different pH values: (**a**) pH = 4.3 (λ = 323.5); (**b**) pH = 5.3 (λ = 326.0); and (**c**) pH = 8.3 (λ = 332.0).

**Figure 6 ijms-20-00416-f006:**
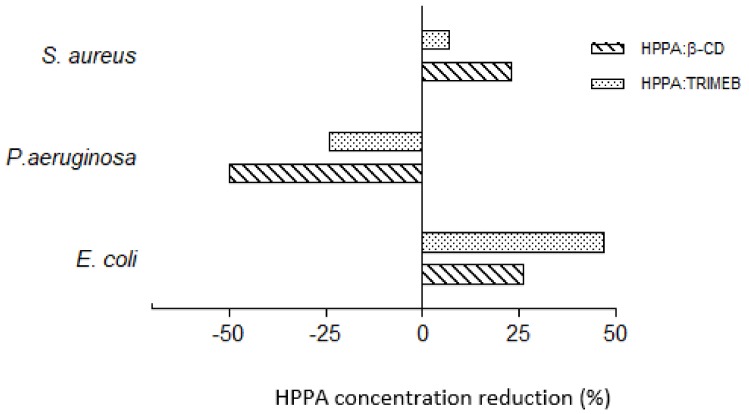
Bacterial growth inhibition: HPPA reduction percentages in KNDs on *E. coli*, *P. aeruginosa*, and *S. aureus* at 24 h of exposure to induce 50% of the bacterial growth inhibition.

**Figure 7 ijms-20-00416-f007:**
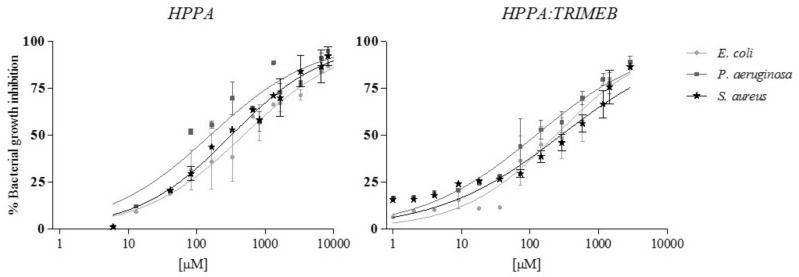
The concentration/response curves of HPPA and HPPA:TRIMEB in the Microbial Susceptibility Test on *E. coli*, *P. aeruginosa*, and *S. aureus*: The trends are from the interpolation of three independent experiments, using GraphPad Prism 5. The results are expressed in μM. The bars represent the standard deviation.

**Figure 8 ijms-20-00416-f008:**
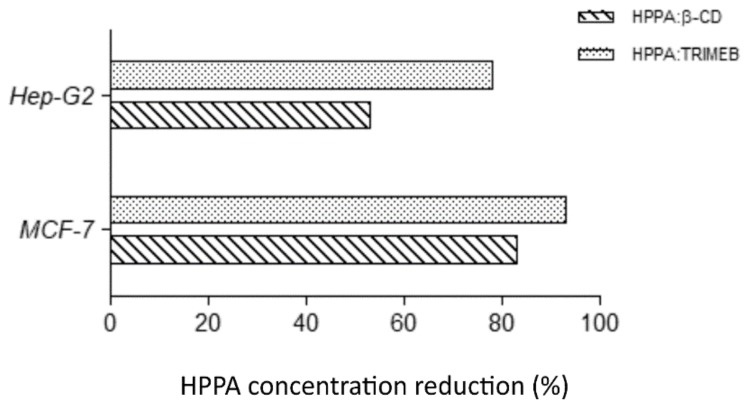
The growth inhibition in tumoral cells: the HPPA reduction percentages in KNDs in Hep-G2 and MCF-7 at 72 h of exposure to induce 50% of the cellular growth inhibition.

**Figure 9 ijms-20-00416-f009:**
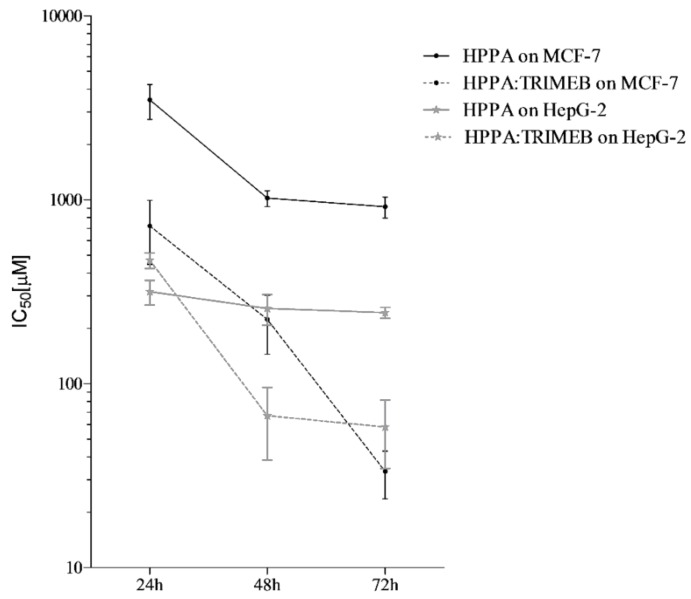
The antitumor activity expressed as IC50 (µM) against MCF-7 and HepG2 cells of HPPA and HPPA:TRIMEB after 24, 48, and 72 h of co-incubation. The trends are from the interpolation of the three independent experiments, using GraphPad Prism 5. The results are expressed in μM. The bars represent the standard deviation.

**Table 1 ijms-20-00416-t001:** The binding constants of HPPA:TRIMEB evaluated by absorbance measurements at different pH values.

pH	Kb (M^−1^) HPPA:TRIMEB
4.3	82.0 ± 37.5
5.3	224.0 ± 18.4
8.3	306.1 ± 74.8

**Table 2 ijms-20-00416-t002:** Antimicrobial activity.

Bacterial Strain	TRIMEB	HPPA	HPPA:TRIMEB
*Escherichia coli*	N.E. up to 3497	473 (329–690)	249 ** (155–399)
*Pseudomonas aeruginosa*		98 (46–210)	122 (87–171)
*Staphilococcus aureus*		314 (240–410)	291 (214–396)

IC_50_ values are expressed in µM with a 95% confidence range (in brackets) after 24 h of exposure. Asterisks highlight significant differences from HPPA (Dunnett’s test: ** *p* < 0.01). N.E. is No Effect.

**Table 3 ijms-20-00416-t003:** IC_50_ values.

Cell Line	t (h)	TRIMEB	HPPA	HPPA:TRIMEB
Hep-G2	24	N.E. up to 1750	360 (290–440)	497 (270–920)
	48		300 (240–370)	59 ** (33–103)
	72		260 (150–470)	57 ** (34–93)
MCF-7	24		3040 (2420–3800)	1025 * (650–1620)
	48		1290 (980–1710)	160 *** (100–250)
	72		750 (580–950)	43 *** (20–90)

IC_50_ values are expressed in µM with a 95% confidence range (in brackets). The asterisks highlight the significant differences from HPPA (Dunnett’s test: * *p* < 0.05; ** *p* < 0.01; *** *p* < 0.0001). N.E. is No effect.

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
