# Peer review of "A New Approach for Improving the Antibacterial and Tumor Cytotoxic Activities of Pipemidic Acid by Including It in Trimethyl-β-cyclodextrin"

_ijms, 2019, doi:10.3390/ijms20020416_

Round 1
Reviewer 1 Report
The article “A new approach for improving the antibacterial and tumor cytotoxic activities of pipemidic acid by including it in trimethyl-β-cyclodextrin” is overall well written.
In my mind, some points could be improved:
* at the beginning of the introduction, some sentences are too long and difficult to understand.
- l33-37: one sentence, with different ideas, and not well expressed.
- l72-76, one sentence, a little long.
* the IR spectra are hard to compare amongst themselves. HPPA IR spectrum is very small. There are not peak picking neither landmarks to view the presumed shifts (and the values) of the NH. Moreover, we should see 2 C=O vibrations for HPPA (Chem. Pharm. Bull. 51 (5) 494-498 (2003 )) and they shouldn’t disappear: the ring C=O should shift due to the break of the intramolecular hydrogen bond. And be careful, I don’t think that Rizzi and al assign as precisely the C-O stretching (l.206).
* -l223, refer to Fig 3. There is only the job plot for the complex …
Finally, I have noted some errors that should be corrected:
- l20 and l25: HPPA instead of HAPPA
- l28, a coma after “ of quinolones”
- l57 bracket forgotten after ref8, and point after ref9
- l77 no dash before “pyrido” and “d” in italic. Add Fig 1after acid
- l99, in Fig 1, add R=Me
- l299 trimethyl
- l338 is less active compared to…
- l349, the legend is not at the right place (but l345)
- l356, I think it is Fig 7 instead of Fig 6
- l366 cyclodextrin
-l387 Sissi C, Palumbo M. The quinolone family: from antibacterial to anticancer agents. Curr Med Chem.-AnticancerAgents 387 2003;3:439-450
-l392 Antibiotics. 2017; 6(26):1-24 should be modified into Antibiotics. 2017; 6(4):26
Author Response
The article “ A new approach for improving the antibacterial and tumor cytotoxic activities of pipemidic acid by including it in trimethyl-β-cyclodextrin” is overall well written.
In my mind, some points could be improved:
* at the beginning of the introduction, some sentences are too long and difficult to understand.
- l33-37: one sentence, with different ideas, and not well expressed.
- l72-76, one sentence, a little long.
R: Sentences were rewritten
* the IR spectra are hard to compare amongst themselves. HPPA IR spectrum is very small. There are not peak picking neither landmarks to view the presumed shifts (and the values) of the NH. Moreover, we should see 2 C=O vibrations for HPPA (Chem. Pharm. Bull. 51 (5) 494-498 (2003 )) and they shouldn’t disappear: the ring C=O should shift due to the break of the intramolecular hydrogen bond. And be careful, I don’t think that Rizzi and al assign as precisely the C-O stretching (l.206).
R: Fig.2 was improved and in the text the following sentence was written:“The FT-IR spectrum of TRIMEB showed prominent absorption bands at 2933 cm-1 (symmetric and asymmetric C-H stretching), 1158 cm-1 (for C-O stretching vibrations), and 1016 cm-1 near the region 1076–1022 cm−1 (C-O-C stretching vibrations), that are in good agreement with the values reported by Rizzi et al. [28]”.
* -l223, refer to Fig 3. There is only the job plot for the complex …
R: Thank you for your suggestions. It was a mistake. New Fig 3, concerning XRD patterns for HPPA, TRIMEB, KND and PM, was added.
Finally, I have noted some errors that should be corrected:
- l20 and l25: HPPA instead of HAPPA
R: Done
- l28, a coma after “ of quinolones”
R: Done
- l57 bracket forgotten after ref8, and point after ref9
R: Done
- l77 no dash before “pyrido” and “d” in italic. Add Fig 1after acid
R: Done
- l99, in Fig 1, add R=Me
R: Done
- l299 trimethyl
R: Done
- l338 is less active compared to…
R: Done
- l349, the legend is not at the right place (but l345)
R: The legend was shifted
- l356, I think it is Fig 7 instead of Fig 6
R: Done
- l366 cyclodextrin
R: Done
-l387 Sissi C, Palumbo M. The quinolone family: from antibacterial to anticancer agents. Curr Med Chem.-AnticancerAgents 387 2003;3:439-450
R: Done
-l392 Antibiotics. 2017; 6(26):1-24 should be modified into Antibiotics. 2017; 6(4):26
R: Done
Reviewer 2 Report
Review comments on the manuscript.
The descriptions of the FT-IR spectra described in the paragraph beginning from L. 205 do not coincide with Fig. 2, particularly about the spectrum of HPPA.
The authors say XRD patterns are shown in Fig. 3. But it doesn’t.
Stain ID of the bacteria used is mandatory, ex. ATCC 12345, WDCM 12345.
It is impossible to judge the manuscript because of the first two problems.
Recommendation: Reject (and possibly resubmit).
Author Response
The descriptions of the FT-IR spectra described in the paragraph beginning from L. 205 do not coincide with Fig. 2, particularly about the spectrum of HPPA.
R: Fig 2 was changed
The authors say XRD patterns are shown in Fig. 3. But it doesn’t.
R: Thank you for your suggestions, it was a mistake. Fig 3 was added
Stain ID of the bacteria used is mandatory, ex. ATCC 12345, WDCM 12345.
R: Bacterial IDs are reported in Introduction section.
Round 2
Reviewer 2 Report
The manuscript is finely reviced.
Recomendation: accept